# Mass Spectrometry Imaging of In Vitro *Cryptosporidium parvum*-Infected Cells and Host Tissue

**DOI:** 10.3390/biom13081200

**Published:** 2023-07-31

**Authors:** Nils H. Anschütz, Stefanie Gerbig, Parviz Ghezellou, Liliana M. R. Silva, Juan Diego Vélez, Carlos R. Hermosilla, Anja Taubert, Bernhard Spengler

**Affiliations:** 1Institute of Inorganic and Analytical Chemistry, Justus Liebig University Giessen, 35392 Giessen, Germany; nils.h.anschuetz@anorg.chemie.uni-giessen.de (N.H.A.); stefanie.gerbig@transmit.de (S.G.); parviz.ghezellou@anorg.chemie.uni-giessen.de (P.G.); 2Institute of Parasitology, Biomedical Research Center Seltersberg, Justus Liebig University Giessen, 35392 Giessen, Germany; liliana.silva@vetmed.uni-giessen.de (L.M.R.S.); juan.velez@vetmed.uni-giessen.de (J.D.V.); carlos.r.hermosilla@vetmed.uni-giessen.de (C.R.H.); anja.taubert@vetmed.uni-giessen.de (A.T.); 3Egas Moniz Center for Interdisciplinary Research (CiiEM), Egas Moniz School of Health & Science, 2829-511 Caparica, Portugal; 4MED—Mediterranean Institute for Agriculture, Environment and Development & CHANGE—Global Change and Sustainability Institute, Institute for Advanced Studies and Research, Universidade de Évora, 7006-554 Évora, Portugal

**Keywords:** *Cryptosporidium parvum*, mass spectrometry imaging, AP-SMALDI

## Abstract

*Cryptosporidium parvum* is a zoonotic-relevant parasite belonging to the phylum Alveolata (subphylum Apicomplexa). One of the most zoonotic-relevant etiologies of cryptosporidiosis is the species *C. parvum*, infecting humans, cattle and wildlife. *C. parvum*-infected intestinal mucosa as well as host cells infected in vitro have not yet been the subject of extensive biochemical investigation. Efficient treatment options or vaccines against cryptosporidiosis are currently not available. Human cryptosporidiosis is currently known as a neglected poverty-related disease (PRD), being potentially fatal in young children or immunocompromised patients. In this study, we used a combination of atmospheric pressure scanning microprobe matrix-assisted laser desorption/ionization (AP-SMALDI) mass spectrometry imaging (MSI) and liquid chromatography-tandem mass spectrometry (HPLC-MS/MS) to determine and locate molecular biomarkers in in vitro *C. parvum*-infected host cells as well as parasitized neonatal calf intestines. Sections of *C. parvum*-infected and non-infected host cell pellets and infected intestines were examined to determine potential biomarkers. Human ileocecal adenocarcinoma cells (HCT-8) were used as a suitable in vitro host cell system. More than a thousand different molecular signals were found in both positive- and negative-ion mode, which were significantly increased in *C. parvum*-infected material. A database search in combination with HPLC-MS/MS experiments was employed for the structural verification of markers. Our results demonstrate some overlap between the identified markers and data obtained from earlier studies on other apicomplexan parasites. Statistically relevant biomarkers were imaged in cell layers of *C. parvum*-infected and non-infected host cells with 5 µm pixel size and in bovine intestinal tissue with 10 µm pixel size. This allowed us to substantiate their relevance once again. Taken together, the present approach delivers novel metabolic insights on neglected cryptosporidiosis affecting mainly children in developing countries.

## 1. Introduction

*Cryptosporidium parvum* belongs to the phylum Alveolata (subphylum Apicomplexa) and is the most relevant zoonotic etiology of human cryptosporidiosis. Additionally to humans, this enteropathogenic parasite infects a wide range of vertebrate hosts, including domestic as well as wild animals [1]. Oral infection with *C. parvum* oocysts occurs through contact with infected people/animals shedding at these infective stages or through drinking contaminated water or eating food washed with contaminated water [2]. *C. parvum* thick-walled oocysts are a highly resistant exogenous stage in the life cycle and are released through the feces of infected humans or animals, thereby leading to new infections upon subsequent ingestion by a new host. Ingested *C. parvum* thick-walled oocysts rupture in the gastrointestinal tract of vertebrate hosts. In this process, known as excystation, four sporozoites are released into the gut lumen. Free-released sporozoites must infect small intestinal epithelial cells (IECs) to achieve further parasite proliferation. In addition to thick-walled oocysts, a second type of thin-walled oocysts develops which hardly resists environmental effects. Nonetheless, these thin-walled oocysts might lead to endogenous autoinfections, probably being the main cause of persistent infection and disease in immunocompromised patients [1]. The epicytoplasmatic intracellular location of *C. parvum* sporozoites with their basal adhesive zones (feeding layers) is another peculiarity of this species when compared to other apicomplexans. Sporozoites within IECs will then undergo asexual merogony resulting in merozoites of the first generation, and these merozoites will again undergo a second merogony producing merozoites of the second merogony. Second-generation merozoites will conduct sexual gamogony, resulting in syngamy and final oocyst production. Afterwards, resistant thick-walled oocysts are excreted from the host through fecal material and the life cycle is completed.

Human cryptosporidiosis still represents a neglected poverty-related disease (PRD) [3]. The most common symptoms of human cryptosporidiosis are abdominal pain, nausea, anorexia, fever and profuse watery diarrhea [1]. The burden of this enteric PRD is currently considerably high in low-income countries [4]. It is a leading source of severe pediatric diarrhea [5]. In already weakened patients, especially young children, the course of the disease can be fatal [1,5]. Diarrheal diseases are responsible for 9% of global child morbidity and mortality [6]. Also, children’s growth can be adversely affected after clinically manifested cryptosporidiosis [4]. Despite intense health consequences, no effective treatments or vaccines against cryptosporidiosis exist [7]. The well-tolerated antiparasitic agent nitazoxanide is the only FDA-approved drug for treating human cryptosporidiosis; nonetheless, it lacks efficacy in immunocompromised patients [7,8,9].

As an obligate fast-replicating parasite, *C. parvum* has reduced metabolic capacities due to a reductive evolutionary process. This minimal metabolic capability is further evidenced in its small genome comprising 9.1 Mb within eight chromosomes when compared to the *Plasmodium* genome of 23 Mb, contained in 14 chromosomes, or to other closely related apicomplexans such as *Toxoplasma*, *Neospora*, *Besnoitia* and *Eimeria* [10,11,12]. Consistently, *C. parvum* lacks a tricarboxylic acid cycle, oxidative phosphorylation, de novo pyrimidine and amino acid and cholesterol biosynthesis [10]. In this regard, it should be noted that the genus *Cryptosporidium* is an extreme example of reductive evolution among apicomplexans [11]. The metabolic possibilities of the parasite are therefore limited and it must use the IECs’ metabolic abilities for successful fast intracellular replication. In this context, it was shown that *C. parvum*-infected host cells experience an upregulation of glycolysis and glutaminolysis [12,13]. Referring to in vitro culture systems, *C. parvum* withstands continuous cultivation and therefore many traditional biochemical methods as well as high-throughput drug screening are very limited [10]. Magnuson et al. used non-imaging MALDI-MS to fingerprint pure *C. parvum* oocysts [14]. The study by Snelling et al. is the first major proteomic study of *C. parvum*, later supplemented by Li et al. [15,16]. Mauzy et al. as well as Matos et al. provided a comprehensive transcriptome of the intracellular stages of *C. parvum* [17,18]. Mass spectrometry imaging (MSI), despite its well-known capabilities, has not yet played a predominant role in studying the metabolic signatures of *C. parvum*-infected host cells in vitro or in infected intestinal tissue. To the best of our knowledge, a comprehensive investigation of *C. parvum*-infected host cells and bovine intestine using MSI has not yet been performed in a biochemical context.

Since the beginning of the millennium, there have been significant improvements in MS instrumentation, and detailed mass analysis is now possible, allowing elemental formulae of compounds to be determined from highly accurate molecular mass values [19,20,21]. Annotations as well as subsequent identifications of analytes are possible. MSI provides the visualization of analyte distributions in tissues and cells. In this context, parasites can be visualized by MSI if infection markers are determined. The size of *C. parvum* oocysts and of the intracellular stages (i.e., sporozoites, trophozoites, meronts and gamonts) is approximately 4–5 μm in diameter [1], requiring MSI methodology with at least 5 μm lateral resolution. Due to improvements in MSI instrumentation, lateral resolutions (image pixel sizes) of 1–2 µm are now available, meaning that assembled or even single *C. parvum*-infected host cells can be examined with this method [22].

In this work, we applied MALDI-MSI and HPLC-MS/MS to investigate the metabolic alterations arising from an infection of *C. parvum* in human intestinal epithelial cells (HCT-8) as well as infected small intestines of neonatal calves to resemble an in vivo scenario as closely as possible. Our study provides the first molecular biomarkers of single *C. parvum*-infected human cells as well as bovine tissue, thereby improving the understanding of the metabolic alterations underlying pathophysiological mechanisms during cryptosporidiosis. Herein, analytes identified through combined MALDI-MSI/HPLC-MS will hopefully help to identify potential novel anti-cryptosporidial drug targets in the future.

## 2. Experimental Section

### 2.1. Parasites

The oocysts of *Cryptosporidium parvum* were obtained from experimentally infected neonatal calves kept at the Institute of Parasitology, Leipzig University, Germany, as previously reported [23]. *C. parvum* strain herein used belongs to the subtype 60-kDa glycoprotein (gp60) IIaA15G2RI and is the most common zoonotic subtype in Germany and in other industrialized countries. Preservation medium of oocysts was composed of sterile phosphate-buffered saline (PBS 1X, pH 7.4; Sigma-Aldrich, St. Louis, MI, USA) supplemented with 100 UI penicillin and 0.1 mg streptomycin/mL (Sigma-Aldrich) at 4 °C. Oocyst stocks were conserved for a maximum of three months to guarantee infectivity of sporulated oocysts. The *C. parvum* oocyst preservation medium was renewed monthly [12].

### 2.2. Cell Culture

Human ileocecal adenocarcinoma cells (HCT-8; ATCC CCL-244™) were maintained at 37 °C and 5% CO_2_ using RPMI 1640 medium (Sigma-Aldrich, St. Louis, MI, USA) supplemented with 0.3 g/L l-glutamine (Sigma-Aldrich, St. Louis, MI, USA), 10% fetal bovine serum (FBS; Biochrom GmbH, Berlin, Germany), 100 UI penicillin and 0.1 mg streptomycin/mL (Sigma-Aldrich, St. Louis, MI, USA). The cell culture medium was changed every other day. Within one experiment, cells from the same passage were used for the infection assays.

### 2.3. Host Cell Infection

HCT-8 cells were seeded at a density of 1 × 10^5^ cells/well on 10 mm round glass coverslips in 24-well plastic microtiter plates (Eppendorf, Hamburg, Germany). When cells reached 60–70% confluence, phase-contrast images were acquired with an inverted microscope (IX81, Olympus, Hamburg, Germany) equipped with a digital camera (XM10, Olympus, Hamburg, Germany) and the total cell number was determined to calculate the infection dose, applying an MOI (multiplicity of infection) of 0.5 (1 sporozoite per 2 host cells), as previously described [12]. Briefly, sporulated *C. parvum* oocysts were pelleted at 5000× *g* for 5 min at 4 °C. Thereafter, sporozoite excystation was induced by supplementation of acidified (pH 2.0) and sterile pre-warmed (37 °C) 1X Hank’s balanced salt solution (HBSS; Sigma-Aldrich) for 10 min at 37 °C. Thereafter, excysted sporozoites were pelleted (5000× *g* for 5 min) and re-suspended in sterile RPMI 1640 cell medium supplemented with 0.3 g/L l-glutamine, 10% FBS, 100 UI penicillin and 0.1 mg streptomycin/mL (all Sigma-Aldrich, St. Louis, MI, USA). Free-released sporozoites were added to HCT-8 monolayers for 3 h, and thereafter cell layers were washed thrice in order to remove free sporozoite and oocyst remnants, and fresh cell culture medium was added.

### 2.4. Preparation of Cell Monolayers and Parasite Pellets

HCT-8 cells were seeded on glass coverslips (10–15 mm, Thermo Fisher Scientific, Dreieich, Germany) and allowed to grow until 60–70% confluency. Cell layers were then infected as described above. At 24 hpi, cell culture medium was removed, and monolayers were washed with sterile PBS 1X before allowing them to dry and freeze at −80 °C until further use. For preparation of parasite pellets, 1 × 10^9^ *C. parvum* oocysts were pelleted (5000× *g*, 5 min) in microcentrifuge tubes (1.5 mL, Eppendorf, Hamburg, Germany). After 2 washes in PBS 1X, the supernatant was removed and the pellet was frozen at −80 °C until further use.

### 2.5. Preparation of C. parvum-Infected Bovine Intestinal Cryo-Sections

Three sections were prepared from each intestinal tissue sample to serve as technical replicates. Both the naturally (*n* = 1) and experimentally (*n* = 2) *C. parvum*-infected intestinal samples were obtained from neonatal calves at the University of Leizig, Germany, as well as the associated control samples. Intestinal tissue samples were embedded in gelatine (8 vol.%) prior to sectioning. The section thickness of the naturally and artificially *C. parvum*-infected intestine sample was 30 µm. The sectioning was performed at −25 °C with a cutting angle of 11° using a Microm Sec35p^®^ blade on a microcryotome (Microm HM 525, Microm International GmbH, part of Thermo Fisher Scientific, Walldorf, Germany). The quality of the sections was checked with an optical microscope (VHX-5000, Keyence, Japan). Subsequently, the sections were stored at −80 °C until AP-SMALDI MSI analysis.

### 2.6. MALDI-MS Sample Preparation

For MALDI measurements in positive-ion mode, 2,5-dihydroxybenzoic acid (DHB, Merck, Darmstadt, Germany) was used as a matrix. The matrix solution was produced by dissolving DHB (30 mg/mL) in 1:1 acetone/water with 0.1% trifluoroacetic acid (TFA, Sigma Aldrich, Steinheim, Germany). The host tissue as well as the cell pellet sections were covered with 100 µL of the matrix solution at a constant flow rate of 10 µL/min using a dedicated pneumatic sprayer (SMALDIPrep, TransMIT GmbH, Giessen, Germany).

### 2.7. Metabolite Extraction

The sample was mixed with 25 µL 0.1% ammonium acetate (Honeywell, Riedel-de Haen, LC-MS Chromasolv^TM^). For cell lysis, the cell pellets were transferred to a potter homogenizer and the intestinal samples to a Mini-Mill PULVERISETTE 23 (Fritsch, Idar-Oberstein, Germany). Metabolites were extracted using methyl tert-butyl ether (MTBE) and methanol as described in the literature [24]. In summary, 100 µL methanol (Sigma Aldrich, St. Louis, MI, USA) and 400 µL MTBE (Sigma Aldrich, St. Louis, MI, USA) were added to this lysate. The mixture was incubated at 4 °C and 900 rpm for 1 h. Subsequently, 200 µL of ice-cold MS-grade water was added and the sample was centrifuged for 10 min at 1000× *g*. The lower aqueous phase was re-extracted, while the upper organic phase was collected. For re-extraction, 200 µL MTBE/methanol/water (4:1.2:1; *v*/*v*/*v*) was added to the aqueous phase. After 1 h of incubation at 4 °C and 900 rpm and centrifugation for 10 min at 1000× *g*, the organic phase was collected again. Organic phases were combined, dried under a nitrogen stream and resuspended in 500 µL acetonitrile/water (60:40; *v*/*v*).

### 2.8. UHPLC-MS/MS Analysis

The gradient settings in HPLC were modified from the literature [25]. Solvent A was acetonitrile/water (60:40) containing 0.1% formic acid and 10 mM ammonium formate (Sigma Aldrich, Germany) and solvent B was isopropanol/acetonitrile/water (90:8:2) containing 0.1% formic acid and 10 mM ammonium formate (Sigma Aldrich, Germany). The elution was performed with a 32 min gradient. The initial condition was 32% of solvent B for 1.5 min. Over a period of 4 min, solvent B was increased to 45%. From 4 to 5 min, solvent B was increased to 52%, from 5 to 8 min to 58%, from 8 to 11 min to 66%, from 11 to 14 min to 70%, from 14 to 18 min to 75% and from 18 to 21 min to 97%. The gradient was kept constant between the 21st and 25th minute. Solvent B was then reduced to 32% and the gradient was held constant for 7 min. Throughout the measurement, the flow rate was maintained at 260 µL/min. Separation was performed on a Dionex UltiMate 3000 RSLC-System (Thermo Fisher Scientific, Dreieich, Germany), using Kinetex C18 (2.1 × 100 mm, 2.6 μm 100 Å particle size) column (Phenomenex, Torrance, CA, USA), coupled to a Q Exactive^TM^ HF-X (Thermo Fisher Scientific, Dreieich, Germany) orbital trapping mass spectrometer.

### 2.9. MALDI-MS(I)

MALDI MS and MALDI MSI experiments were carried out using an AP-SMALDI^5^ AF (TransMIT GmbH, Giessen, Germany) imaging system (pixel size: ≥5 µm) coupled to a Q Exactive^TM^ HF (Thermo Fisher Scientific (Bremen) GmbH, Germany) orbital trapping mass spectrometer (mass resolution was set to R = 240,000 @ *m*/*z* 200). For internal calibration with DHB as a matrix, the lock mass *m*/*z* 716.12462 (corresponding to [5DHB−4H_2_O+NH_4_]^+^) was set. In case of pNA, *m*/*z* 273.06238 (corresponding to [2pNA−H_2_−H]^−^) was used. In positive-ion mode the mass range was set to *m*/*z* 300–1200 and in negative-ion mode *m*/*z* 250–1000. In the case of the cell pellet sections, a pattern of 50 × 50 pixels was measured with a step size of 10 μm. Due to a larger (defocused) laser spot area at 10 µm step size, higher signal intensities were obtained. In addition, the Full Pixel mode of the ion source was used for ablation of the entire 10 µm × 10 µm area by a meandering movement [26].

### 2.10. Data Processing

The software Mirion (TransMIT GmbH, Giessen, Germany), together with the Perseus software platform (MPI of Biochemistry, Martinsried, Germany), was used to find potential biomarkers in the MS data of the cell pellets [27]. The utilized procedure was based on the published literature [28,29]. Using Mirion [30], all MALDI MS measurements of the cell pellets were merged and analyzed together. A list of all *m*/*z* values with a pixel coverage ≥ 0.5% was exported, the corresponding deviations (±5 ppm) were calculated and this modified list was compared to all single-cell measurements. The results were then loaded in Perseus, the datasets were classified into infected and non-infected host cells/intestinal tissues. Data were normalized by dividing the intensity values by their sum and afterwards by using the Z-score (median, for hierarchical clustering). Multiple-sample tests were carried out (ANOVA; permutation-based false discovery rate, FDR = 0.05, number of randomizations = 250). The invalid values were filtered out and post hoc tests were performed with the remaining values (visualized using hierarchical clusters). Annotations of the *m*/*z* values found by Perseus were provided by LipidMaps database. These accurate mass values and their deviations (±5 ppm) were used to create a mass list and utilized as an inclusion list for LC-MS/MS analysis. The MS raw files were renamed by the LipidMatch RenamingTool. Finally, lipids were identified using LipidMatch Flow [27].

## 3. Results and Discussion

*C. parvum*-infected HCT-8 as well as infected bovine neonatal intestinal mucosa were here used as model systems to be as close as possible to in vivo situation where exclusively IECs serve as suitable host cells. Figure 1 illustrates the general workflow. *C. parvum* oocysts were obtained from experimentally infected neonatal calves. Subsequently, oocysts were used to isolate sporozoites to generate cell pellets and infected monolayers. Additionally, *C. parvum*-infected intestine samples were obtained from diseased neonatal calves. Sections from infected and control cell pellets were used to determine statistically significant biomarkers. These biomarkers were then visualized using MALDI-MSI in monolayers and sections of artificially and/or naturally infected small intestinal tissues.

### 3.1. MALDI-MSI Measurements of Cell Pellets

The HCT-8 cell line is the most widely used in vitro system for *C*. *parvum*. studies. Cell pellets were sliced into consecutive thin sections to provide a homogeneous sample pattern for the MALDI-MS experiments and facilitate statistical evaluability. Figure 2 illustrates the chosen experimental approach. Different HCT-8 (*n* = 3) cell sections were measured, and the MS ion at *m*/*z* 530.3212 that was found to be a statistically relevant infection marker is displayed using the red color channel (Figure 2A,B). Figure 2A shows two images with a size of 50 × 50 pixels. Each analysis of the cell pellets was performed with a step size of 10 µm. The larger step sizes, along with a defocusing of the laser beam, led to larger spot sizes, resulting in a larger amount of sample material being ablated and ionized by the laser beam. Still, no satisfactory results could be achieved in standard “spot mode”. Therefore, measurements of the cell pellets were carried out with a signal-enhancing mode of the ion source, the Full Pixel mode. Here, a larger number of laser pulses are used to ablate the pixel area by a meandering movement [26], while in spot mode, laser pulses are applied only to a single spot in the center of the pixel. As a result, more material is removed from the area of the pixel as well as from the depth of the sample. The Full Pixel mode thus leads to an increase in signal intensities and an improved limit of detection. In contrast, a higher lateral resolution was required for the monolayers and the host tissue (step size and laser focus diameter 5 µm), due to the small size of *C. parvum* stages. Therefore, the spot mode with a higher number of pulses was used for this purpose.

The left part of Figure 2A illustrates an infected section, and the right part the corresponding control sample. The two samples were placed side by side on a sample holder, sprayed with matrix and measured subsequently. This ensured that the experimental conditions were identical. Signal intensities can therefore be interpreted as being quantitative on a relative scale. After that, another two infected and control pairs of technical replicates were measured in order to obtain triplicate measurements (Figure 2B). To exclude possible errors in sample preparation (e.g., sectioning) or possible heterogeneities within the cell pellets, technical replicates were measured. The same conditions were used for the preparation and storage of the samples. The Mirion [30] and Perseus [25] software platforms were used to determine potential biomarkers within the cell sections. The corresponding signals from the preceding step were filtered and non-fitting signals were rejected from the list. In the last step, post hoc tests were performed with the remaining signals (visualized using hierarchical clustering, see Figure 2C). The metabolism of *C. parvum* is known to be highly active [28], and a large number of differences in ion signals between control and infected samples are to be expected. For the in vitro HCT-8 model system, 1114 marker signals were found to be significantly upregulated in cases of infection in positive-ion mode (see Appendix A) and 1118 marker signals in negative-ion mode (see Appendix A). In the case of downregulation, 1330 (see Appendix A) and 1386 marker signals (see Appendix A) were detected in positive- and negative-ion mode, respectively.

### 3.2. Annotation and Verification of Markers

The annotation of previously identified markers was performed using the LIPID MAPS database [31]. Lipids are basic components of the structural and functional categories of all cells. In cell membranes, they separate biofunctional areas and are involved in accomplishing various aspects of signal transmission. In the case of MALDI measurements of cell pellets and their following statistical evaluation, all *m*/*z* signals were taken into account. Due to this approach, the focus was exclusively on lipids. This is also reflected in the extraction process for the HPLC-MS/MS measurements and the database used. As a result, not every signal that was initially recognized by MALDI as a significant marker can subsequently be annotated or identified. In order to obtain meaningful annotations, the LIPID MAPS search criteria were selected to include appropriate ion adducts with the corresponding polarity. Only singly charged species were selected. In positive-ion mode, ions of types [M+H]^+^, [M+H-H_2_O]^+^, [M+Na]^+^, [M+NH_4_]^+^ and [M+K]^+^, and [M-H]^−^, [M+Cl]^−^, [M+HCOO]^−^, [M+OAc]^−^ and [M-CH_3_]^−^ in negative-ion mode, were taken into account. In LIPID MAPS, a mass tolerance of *m*/*z* ±0.05 was chosen and afterwards, values with a calculated deviation of more than 5 ppm between the theoretical and measured values were discarded. Annotations deviating from the *m*/*z* value by more than 1 ppm were discarded if several different annotations were found for the same *m*/*z* value. The corresponding results are illustrated in Figure 3. Because more than one annotation can meet the criteria described above, the number of annotations is larger than the number of *m*/*z* values. A total of 3221 annotations were determined for the positive-ion mode (for 880 different *m*/*z* values, see Appendix A) and 3059 annotations for the negative-ion mode (for 762 different *m*/*z* values, see Appendix A). In the case of downregulation, 2333 annotations (721 different *m*/*z* values, see Appendix A) were determined for the positive-ion mode and 1036 (380 different *m*/*z* values) annotations for the negative-ion mode (see Appendix A). Figure 3 shows the fractions of signal numbers (in [%]) of all annotated lipid categories (A1, B1, C1 and D1) as well as the various subclasses of the particularly prominent glycerophospholipids (A2, B2, C2 and D2). While parts A and B depict the results for the positive-ion mode, parts C and D show the results for the negative-ion mode. In the positive-ion mode, the proportions of lipid categories in the infected and control samples only minimally changed. Within the group of glycerophospholipids, the proportions of lipid classes slightly changed. The *C. parvum*-infected samples showed a larger number of phosphatidylserines (PSs) and phosphatidylinositols (PIs). The relative amount of phosphatidylcholines (PCs), phosphatidylethanolamines (PEs) and especially lysolipids (Lyso-) was found to be decreased in *C. parvum*-infected samples. In the negative-ion mode, the proportions of lipid classes changed significantly. The proportion of glycerophospholipids increased with parasitic infection. On the class level, the proportions of PC and PE decreased, while the ones of PI, PA and lysolipids increased.

Inclusion lists for the HPLC-MS/MS measurements were generated using the annotations obtained with LIPID MAPS. In addition, high-resolution full MS and MS/MS spectra were recorded. LipidMatch Flow in combination with ProteoWizard (MSConvertGUI) and Mzmine were used to identify the molecular markers already detected [32,33,34]. The corresponding settings for LipidMatch Flow can be found in Appendix A.

In total, 37 of the annotations for upregulated lipids in *C. parvum*-infected samples in the positive-ion mode were confirmed by the MS/MS experiments (see Appendix A). The PC lipid class, a major structural component of membrane bilayers, was highly abundant. PC is known to be the dominant phospholipid class for *C. parvum* [35]. It is also significantly increased in infections with other apicomplexans, e.g., *Toxoplasma gondii* [36]. It is also known that PC is the most abundant phospholipid class in *T. gondii* and *Plasmodium falciparum* membranes [37,38]. PC synthesis is crucial for the replication of *T. gondii* tachyzoites and *P. falciparum* blood and liver stages [39,40,41,42]. In infected samples, 15 annotations of downregulated lipids were confirmed in positive-ion mode (see Appendix A). In negative-ion mode, 15 identifications were obtained from the previously annotated biomarkers in the case of upregulation (see Appendix A). In this context, 20 identifications could be obtained for downregulation (see Appendix A).

Irrespective of the ion mode, it has also been shown that this apicomplexan infection has an influence on the composition of the lysolipids. They are known for being ubiquitous intermediates in a variety of metabolic and signaling pathways in eukaryotic cells [43]. In the case of *P. falciparum*, there is evidence that the parasite lysophospholipase is used to generate phosphocholine for parasite PC synthesis from lysolipids already available in the parasitized host [44]. This is in agreement with our observation of a change in PC and lysolipid abundances in *C. parvum*-infected host cells and intestinal mucosa.

### 3.3. Mass Spectrometry Imaging of C. parvum-Infected Cell Layers and Intestinal Tissue at High Lateral Resolution

In positive-ion mode, 38 upregulated signals related to *C. parvum* infection and 16 downregulated signals were identified in both HCT-8 pellets and host tissue (see Appendix A). Visualization of the small parasites (3–5 µm) attached to the cells requires a lateral resolution of at least 5 µm for the MSI of monolayers and host tissue. In each panel of Figure 4, *C. parvum*-infected HCT-8 layers are shown at the bottom, and control samples at the top. Images were obtained in positive-ion mode. The green channel represents the total ion count (TIC) in the two images. The red channel shows the distribution of molecular markers obtained by LC-MS/MS analysis (mass tolerance of ±5 ppm) and identified as infection markers. As expected, these signals were found exclusively or with significantly higher signal intensity in *C. parvum*-infected cell layers (bottom side of each panel) than in the control samples. In Figure 4A, the distribution of an infection-specific signal at *m*/*z* 766.5720 assigned to [plasmanyl-PC(O-16:1/20:4)+H]^+^ is shown, and Figure 4B displays the distribution of the signal at *m*/*z* 504.3036, assigned to [LPE(20:3)+H]^+^. As mentioned earlier, the signal increase in the lipid class of the PC is typical of infection with apicomplexan parasites. LPE is part of cell membranes, which play an essential role in the activation of other enzymes and cell-mediated cell signaling [45].

In addition to cell monolayer measurements, cryo-sections of *C. parvum*-infected neonatal bovine intestinal tissue were analyzed. Figure 5A,B show MS images of a neonatal bovine intestine section. In Figure 5A, the green channel represents an ion channel at *m*/*z* 756.5513, used for visualization purposes only and indicating the complete intestine section. The distribution of two infection-specific signals at *m*/*z* 504.3036, identified as [LPE(20:3)+H]^+^, and *m*/*z* 530.3058, identified as [LPE(22:4)+H]^+^, are shown in red and blue. It can be seen that the markers for the infection only occur in the inner or luminal area of the small intestine (outlined in white). This is the area where *C. parvum* replication takes place, resulting in trophozoite, meront, gamont and oocyst stages. To substantiate this fact, Figure 5B shows exclusively the infection biomarkers from Figure 5A. In Figure 5C, the corresponding optical image of the whole intestine section can be seen. The infected area is outlined in red.

Finally, experimentally *C. parvum*-infected neonatal calf intestinal samples were examined by MS imaging. The samples were prepared in such a way that the inner part of the intestine, and therefore the site of infection, corresponded exclusively to the right part of the section. Figure 6A shows an MS image of the measured area, the corresponding optical images can be found in B (entire section) and C (measured area). It can be clearly seen that the biomarkers for the infection were exclusively detected in the right border of the section, which is where infective *C. parvum* sporozoites were applied. Panels A2 to A6 represent another 10 selected marker signals for infection.

The identified biomarkers as determined for *C. parvum* were compared to biomarkers of *T. gondii-*, *Besnoitia besnoiti-* and *Neospora caninum*-infected host cells, as previously published [28,29]. These parasites all belong to the taxonomic subphylum Apicomplexa and are thus suspected to show metabolic similarities to *C. parvum*. A comparison is shown in Appendix A. PI (36:1) as [M-H]^−^ was identified as a marker for host cell infection by *C. parvum* as well as by *T. gondii*, *B. besnoiti* and *N. caninum*. Phosphatidylinositol is present in cell membranes, playing an important role in anchoring proteins to the membrane. For *T. gondii*, it has been shown that PIs play roles in the parasite’s transition between its stages, i.e., tachyzoites and bradyzoites, and therefore are essential for establishing persistent infections in mammalian hosts, including humans [46]. To assess the metabolite profiles associated with infection, Hublin et al. performed a gas-chromatography-MS-based metabolomic analysis (mass range *m*/*z* 45–600) of fecal samples from experimentally infected mice. The seven lipids mentioned in the study, which were associated with a significant change in concentration, do not show any intersection with our results. In addition to different ionization techniques and the restriction to the positive-ion mode, the different selected mass ranges limit a possible intersection [47].

## 4. Conclusions

To the best of our knowledge, this is the first MSI study examining *C. parvum*-infected human host cells and bovine intestinal tissue at the molecular level to be as close as possible to an in vivo situation. By comparing technical triplicates of infected and non-infected sections of cell pellet samples, it was possible to find lipid biomarkers for infection with the parasite *C. parvum*, which belongs to the subphylum Apicomplexa. AP-SMALDI MS and MSI were used for describing obligate intracellular infections. The MS measurements were carried out in positive- and negative-ion modes, and the results were interpreted with software including statistical tools. With this method, 2232 ion signals were detected in *C. parvum*-infected samples, and were significantly different from those detected in the control ones. Using databases and structure elucidation based on MS/MS experiments, it was possible to structurally identify 54 biomarkers, while the others remained structurally undetermined. The obtained data were compared with already published data on other apicomplexan parasites (*T. gondii*, *B. besnoiti* and *N. caninum*). PI (36:1) was identified as a common marker for infection with all three mentioned parasites. For MSI experiments, cell layers as well as natural and experimentally infected host intestinal tissues were analyzed and previously found markers were imaged. This allowed infected cells or host intestinal tissue to be distinguished from uninfected cells and healthy host tissue. In this way, the parasitic infection could be made visible at a 5 to 10 µm lateral resolution. The presented research will be the basis for future detailed investigations on *C. parvum*’s modulation of host cell metabolism. Such novel lipid data then will aid in uncovering alternative metabolic pathways, providing additional drug targets against *C. parvum* and other apicomplexan parasites of veterinary and public health importance.

## Figures and Tables

**Figure 1 biomolecules-13-01200-f001:**
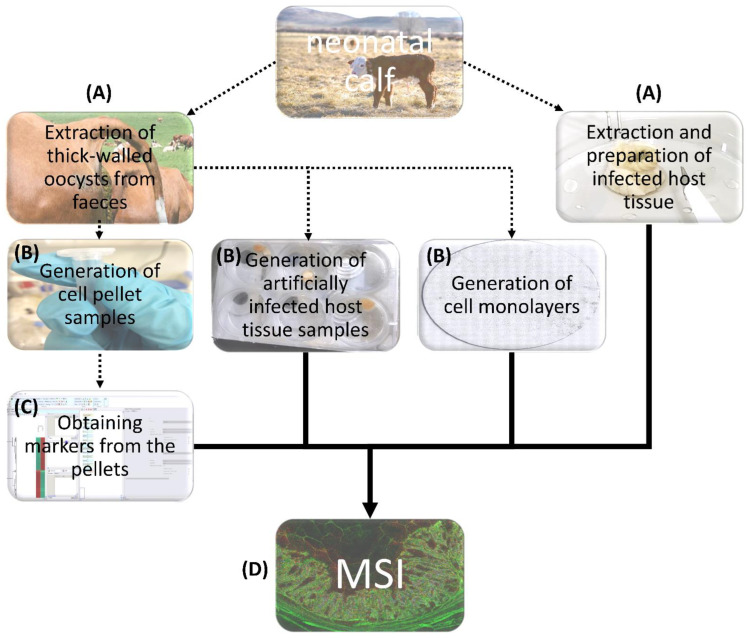
General workflow. Cryptosporidium parvum oocysts, infected intestine and non-infected intestine were obtained from neonatal calves (**A**). These thick-walled oocysts were then used to generate cell pellets, monolayers and artificially infected intestinal tissues (**B**). Sections from cell pellets were used to obtain statistically significant markers (**C**). Markers were visualized with the help of MALDI-MSI in monolayers and host intestinal tissues (**D**).

**Figure 2 biomolecules-13-01200-f002:**
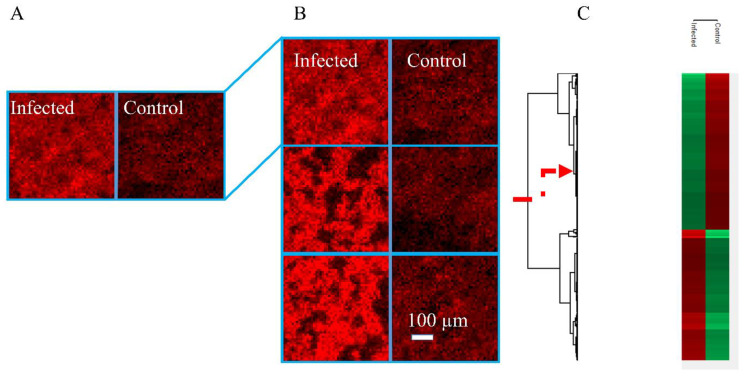
Workflow for determination of biomarkers: (**A**) The left part illustrates an infected section, and the right part the corresponding control sample. Upregulated infection marker LPE (22:4) as [M+H]^+^, *m*/*z* 530.3212 ± 5 ppm, in red, each tile is 50 × 50 pixels with a step size of 10 µm. (**B**) Three technical replicates; LPE (22:4) as [M+H]^+^, *m*/*z* 530.3212 ± 5 ppm, each tile is 50 × 50 pixels with a step size of 10 µm. (**C**) Segment of a heat map generated with Perseus. The color code of the column indicates whether it is a corresponding marker. Red means the signal is significantly increased compared to the other sample group. The red arrow indicates that the visualized marker is one of many markers within the heatmap.

**Figure 3 biomolecules-13-01200-f003:**
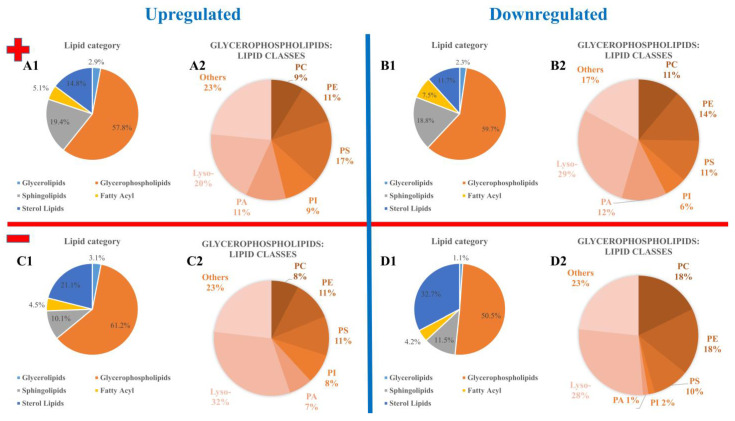
Overview of the annotated lipids, showing abundances of categories of detected HCT-8 markers (fractions of signal numbers (in [%])): (**A**,**B**) positive-ion mode; (**C**,**D**) negative-ion mode; (**A**,**C**) upregulated lipids; (**B**,**D**) downregulated lipids—(**1**) lipid categories, (**2**) detected lipid classes within the glycerophospholipids category.

**Figure 4 biomolecules-13-01200-f004:**
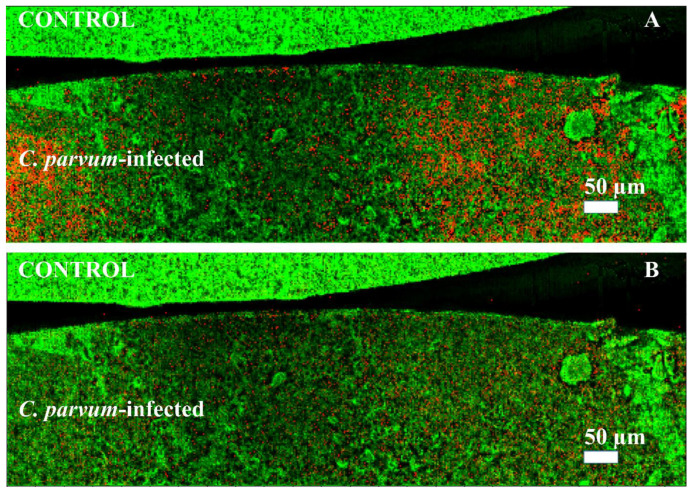
Visualization of biomarkers in cell monolayers: MALDI MSI measurements of cell monolayers in positive-ion mode, measured with 5 µm laser focus diameter and step size. The green channel in both images represents the TIC, used for visualization purposes only. The red channel shows the distribution of two different infection markers (±5 ppm mass tolerance) identified earlier by LC-MS/MS measurements of cell pellets or host tissue. (**A**) Infection marker signal at *m*/*z* 766.5720, identified as plasmanyl-PC(O-16:1/20:4) as [M+H]^+^, (**B**) infection marker signal at *m*/*z* 504.3036, identified as LPE(20:3) as [M+H]^+^.

**Figure 5 biomolecules-13-01200-f005:**
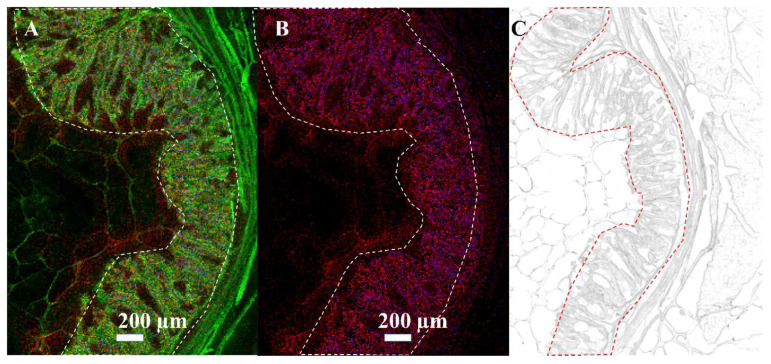
Visualization of infection biomarkers in host tissue: MALDI MSI measurements of *C. parvum*-infected neonatal bovine intestine in positive-ion mode, measured with 10 µm laser focus diameter and step size in Full Pixel mode. The red and blue channels show the distribution of two infection markers (±5 ppm mass tolerance) identified earlier by LC-MS/MS measurements of cell pellets or host tissue. (**A**) The green channel represents an ion signal at *m*/*z* 756.5513, used for visualization purposes only. Infection marker signal at *m*/*z* 504.3036, identified as LPE (20:3) as [M+H]^+^ in red, and *m*/*z* 530.3058, identified as LPE (22:4) as [M+H]^+^ in blue. (**B**) Infection marker signal at *m*/*z* 504.3036, identified as LPE (20:3) as [M+H]^+^ in red and *m*/*z* 530.3058, identified as LPE (22:4) as [M+H]^+^ in blue. (**C**) Optical image of the whole intestine section. The area of infection (wrinkles and villi) is outlined in red.

**Figure 6 biomolecules-13-01200-f006:**
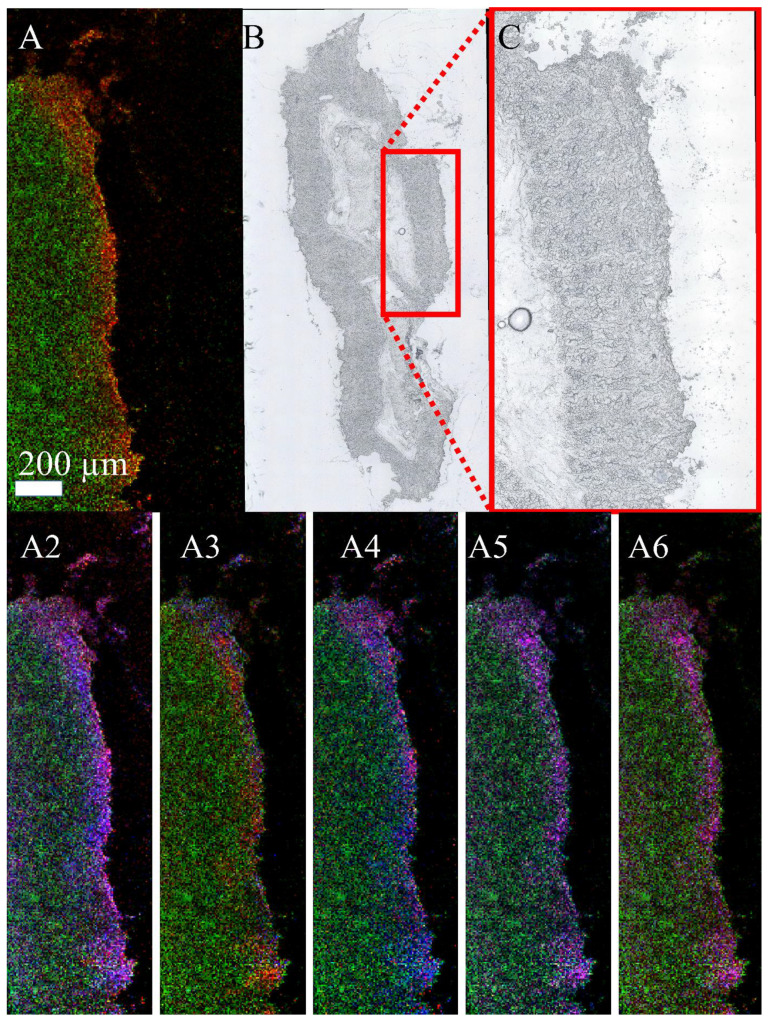
Visualization of biomarkers in experimentally *C. parvum*-infected neonatal bovine intestinal tissue: MALDI MSI measurements of bovine intestine in positive-ion mode, infected with *C. parvum* and measured with 12 µm laser focus diameter and step size. The green channel represents an ion signal at *m*/*z* 726.5603, used for visualization purposes only. The red and the blue channel show the distribution of two infection markers (±5 ppm mass tolerance) identified earlier by LC-MS/MS measurements of cell pellets. (**A**) Infection marker signal at *m*/*z* 504.3057 (red), identified as LPE (20:3) as [M+H]^+^, (**B**) optical image of the whole intestine section, (**C**) zoomed optical image of the measured area. (**A2**–**A6**) confirm the interpretation of A for another ten biomarkers. These ten biomarkers were always found in the same area and their distribution was congruent with A. (**A2**) Infection marker signals at *m*/*z* 532.3371 (red) and 558.2955 (blue), annotated by Lipid Maps as LPC (17:0) as [M+Na]^+^ (further annotations possible) and LPC (18:2) as [M+K]^+^ (further annotations possible); (**A3**) infection marker signals at *m*/*z* 540.3059 (red), annotated by Lipid Maps as LPC (18:3) as [M+Na]^+^ (further annotations possible) and 576.3282 annotated by Lipid Maps as LPS (20:0) as [M+Na]^+^ in blue (further annotations possible); (**A4**) infection marker signals at *m*/*z* 640.3220 (red) annotated by Lipid Maps as PS (24:3) as [M+Na]^+^ (further annotations possible) and *m*/*z* 618.3399 (blue) annotated by Lipid Maps as PS (24:3) as [M+H]^+^; (**A5**) infection marker signals at *m*/*z* 656.3557 (red) annotated by Lipid Maps as PS (27:5) as [M+H]^+^ and *m*/*z* 657.3591 (blue) annotated by Lipid Maps as PI (21:0) as [M+H]^+^; (**A6**) infection marker signals at *m*/*z* 521.3428 (red) annotated by Lipid Maps as DG (25:2;O2) as [M+Na]^+^ and *m*/*z* 520.3394 (blue) annotated by Lipid Maps as LPC (18:2) as [M+H]^+^.

## Data Availability

The data presented in this study are available on request from the corresponding author.

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
