# Peer review of "Mass Spectrometry Imaging of In Vitro Cryptosporidium parvum-Infected Cells and Host Tissue"

_biomolecules, 2023, doi:10.3390/biom13081200_

Round 1

Reviewer 1 Report

This manuscript presents an interesting study focused on identifying molecular biomarkers of infection in Cryptosporidium parvum, a parasite that affects both humans and cattle globally. The authors employ a high-throughput approach using AP-SMALDI, MSI, and HPLC-MS/MS techniques to analyze isolated C. parvum cells, in vitro infected host cells, and parasitized neonatal calf intestine. The results demonstrate significant increases in molecular signals specific to C. parvum-infected materials, with some overlap observed with markers identified in previous studies on related parasites. The authors highlight the unique life cycle and genome of C. parvum within the Apicomplexa group. Nevertheless, some papers on MALDI - C. parvum are not mentioned/discussed. The study's utilization of improved MSI instrumentation and enhanced lateral resolution enables the analysis of C. parvum cells, providing insights into the identification of distinct features and genes that may contribute to the development of new treatments or vaccines.

Specific comments:

Figure 1:

Including labels outside the figure would enhance the clarity and visibility of the image, facilitating the communication of the key points. Additionally, incorporating round connecting lines would improve the overall flow of the figure.

References:

To ensure uniformity, it is advisable to adhere to the MDPI biomolecules reference format. In the manuscript, some references have capitalized titles; some have full names and other abbreviations. Furthermore, scientific names should be in italics to conform to the standard formatting guidelines.

Author Response

Please see the attached cover letter for our response to reviewer comments.

Reviewer 2 Report

The manuscript (biomolecules-2476677) entitled "Mass spectrometry imaging of in vitro Cryptosporidium parvum-infected cells and host tissue" is a very interesting article that addresses the important issue of obtaining lipidomic data that are critical for the evaluation and identification of biomarkers of Cryptosporidium infection. The most important of these assays have already been used in AP-SMALDI, MSI and HPLC MS /MS. These methods are interesting tools for assessing the infectivity of intestinal mucosa infected with C. parvum as well as parasitic intestinal host cells of newborn calves infected in vitro.

Cryptosporidium is the leading cause of diarrheal disease in patients in resource-poor countries around the world. After ingestion of the infectious oocysts of the parasite, motile sporozoites develop and invade the epithelial cells of the small intestine, where they develop in an intracellular but extra cytoplasmic niche. Cryptosporidium completes its complex life cycle in a single host, with both asexual and sexual stages present in the intestine. Multiplication of the parasite and the resulting immune response contribute to the development of severe watery diarrhea in infected individuals. Currently, there is no vaccine and only one drug (nitazoxanide) has limited efficacy in the most susceptible individuals. The results of this work are of particular importance for research because Cryptosporidium is a facultative epicellular apicomplexan that can multiply in a host cell-free environment.

Comments:

the format of the presentation of the results in the appendix (supplementary) in the format WORD is incomprehensible to me. Excel files with separate sheets are preferable. The results obtained by him are very poorly described, but this is understandable, assuming that this is the first such analysis, which will be used for further possible experiments that will allow a deeper and more targeted analysis.

The work is scientifically sound and worthy of consideration for publication. The experimental design is correct; the methods are correctly described. The statistical analysis is correct.

Author Response

(The authors gave the same response as above.)

Reviewer 3 Report

The authors present data describing biochemical changes in Cryptosporidium and Cryptosporidium-infected host cells. The authors concentrate mainly on changes in phospholipids, particularly the phosphatidylcholine class, which is interesting and does add to the literature on the role of this molecule and the disease scenario. However, the data has much more information on several other biochemical changes that could have been developed and added considerably to the impact of the study.

The study is clearly described and the methodology of high quality.      

Specific comments:

Line 84: This statement is not exactly true there are several publications detailing comprehensive biochemical investigations of Cryptosporidium and infected host cells. [Schroeder et al., J Protozool 1999, 85(2):213; Magnuson et al., App Env Microbiol 2000, 66(11): https://doi.org/10.1128/AEM.66.11.4720-4724.2000; Snelling et al., Mol Cell Proteomics 2007, 6(2): 346; Mauzy et al., PLoS One 2012, 7(3):e31715; Lippuner et al., Int J Parasitol 2018, 48(6):413; Liu et al., Frontiers in Microbiol 2018, 9:1409; Matos et al., Sci Rep 2019, 9:7856; Li et al., Parasites Vectors 2021, 14:608; Sun et al., Parasites and Vectors 2022, 15:441]. The authors data extends the current MS data on the parasite during the infection.

Line 443: The phylogenetic classification of the Apicomplexa is mostly based upon morphological characteristics (the intracellular location but extracytoplasmic nature) not on biochemical characteristics, where the Cryptosporidium are more like the Gregarines. Hence not sure this sentence is meaningful.

Line 448: The Toxoplasma tachyzoite-bradyzoite transition does not have a comparison in Cryptosporidium life cycle?

Ng-Hublin’s group at Murdoch Univ (Hublin-Ng et al., 2013 PLoS One 8(10):e77803) has detailed lipid changes during infection by Cryptosporidium, but not mention of this work is included in this study. The authors need to at least acknowledge and compare their results with this study.

Author Response

(The authors gave the same response as above.)
